# Relationship between the Crystal Structure and Tuberculostatic Activity of Some 2-Amidinothiosemicarbazone Derivatives of Pyridine

**DOI:** 10.3390/ma15010349

**Published:** 2022-01-04

**Authors:** Katarzyna Gobis, Małgorzata Szczesio, Andrzej Olczak, Tomasz Pawlak, Ewa Augustynowicz-Kopeć, Malwina Krause, Marek L. Główka

**Affiliations:** 1Department of Organic Chemistry, Medical University of Gdańsk, 107 Gen. Hallera Av., 80-438 Gdansk, Poland; malwina.krause@gumed.edu.pl; 2Institute of General and Ecological Chemistry, Faculty of Chemistry, Lodz University of Technology, Zeromskiego 116, 90-924 Lodz, Poland; malgorzata.szczesio@p.lodz.pl (M.S.); andrzej.olczak@p.lodz.pl (A.O.); marek.glowka@p.lodz.pl (M.L.G.); 3Centre of Molecular and Macromolecular Studies, Polish Academy of Science, 112 Sienkiewicza, 90-363 Lodz, Poland; tpawlak@cbmm.lodz.pl; 4Department of Microbiology, National Tuberculosis and Lung Diseases Research Institute, 26 Płocka Str., 01-138 Warsaw, Poland; e.kopec@igichp.edu.pl

**Keywords:** tuberculostatic activity, SAR-analysis, 2-amidinothiosemicarbazones, pyridine, crystal state X-ray, ^15^N NMR study

## Abstract

Tuberculosis remains one of the most common diseases affecting developing countries due to difficult living conditions, the rapidly increasing resistance of *M. tuberculosis* strains and the small number of effective anti-tuberculosis drugs. This study concerns the relationship between molecular structure observed in a solid-state by X-ray diffraction and the ^15^N NMR of a group of pyridine derivatives, from which promising activity against *M. tuberculosis* was reported earlier. It was found that the compounds exist in two tautomeric forms: neutral and zwitterionic. The latter form forced the molecules to adopt a stable, unique, flat frame due to conjugation and the intramolecular hydrogen bond system. As the compounds exist in a zwitterionic form in the crystal state generally showing higher activity against tuberculosis, it may indicate that this geometry of molecules is the “active” form.

## 1. Introduction

The widespread use of antibiotics in intensive livestock farming, the subsequent penetration of drugs and their metabolites into groundwater, and excessively and hastily prescribing antibiotics resulted in the emergence of many bacterial strains characterized by drug resistance [1]. Increasing drug resistance among species of pathogenic bacteria poses a significant problem for the treatment of infectious diseases. One such disease is tuberculosis, caused by *Mycobacterium tuberculosis* [2]. According to the World Health Organization, 10 million people fell ill with tuberculosis in 2019 (including 1.2 million children under the age of 14), of which 1.4 million died. Five hundred thousand cases were resistant to rifampicin—one of the first-line drugs—and 78% were resistant to other tuberculostatics. Unfortunately, only one in three patients entered treatment [3].

One of the main reasons for the continued spread of TB (tuberculosis) is the narrow range of effective drugs. Their number remains limited due to difficulties associated with searching for new and active substances, despite several findings in structure–activity relationship studies. The difficulties in efficient implementation are the slow replication of mycobacteria, that *M. tuberculosis* belongs to the third group of pathogens according to the biosafety level (BSL), the shortage of scientific centers authorized to conduct research on *M. tuberculosis* strains, difficulties in mapping the specific environment inside the host of mycobacteria in animal models, the not fully understood biology of mycobacteria in latent form, and the hindered penetration of chemotherapeutic agents through the cell wall [4,5,6].

The study described below is a continuation of our previous studies on amidrazone derivatives exhibiting high anti-tuberculosis activity [7,8,9,10,11,12]. The previous studies allowed us to formulate a hypothesis that the planarity of these molecules being isoniazid analogues (INH) is a significant prerequisite for their tuberculostatic activity. Among others, *S*-methyl and *S*,*S*-dimethylamino(pyridin-2-ylmethylene)carbono-hydrazonodithioates were synthesized. These compounds showed high tuberculostatic activity in vitro with a minimum inhibitory concentration (MIC ) value of 3.13 µg/mL. Crystallographic studies were carried out for a flat structure due to conjugation and the formation of intramolecular hydrogen bonds (Figure 1) [12,13].

The same conditions are also met by some 4-substituted picolinonitrile derivatives obtained in the course of this study, having an amidrazone function in the C-2 position of the pyridine ring (Figure 2). For some of them, we obtained very promising activity against *M. tuberculosis* strains with MIC values of 0.4 µg/mL [14]. In this research, we try to explain the relationship between their activity and the structural features of their molecules. For example, an interesting property of the tested amidrazone derivatives is their presence in two tautomeric forms, which results in different conformations and different intramolecular hydrogen bonds. As the molecular mechanism of tuberculostatic activity for the studied compounds is unknown, we tried to find any relationship among more active compounds; however, their number is limited.

## 2. Materials and Methods

### 2.1. Synthesis

All materials and solvents were of analytical reagent grade (Sigma-Aldrich-Merck KGaA, Darmstadt, Germany). Thin-layer chromatography was performed on Merck silica gel 60F_254_ plates and visualized with UV light. The results of elemental analyses (%C, H, N) for all of the obtained compounds were in agreement with the calculated values that were within the ± 0.4%range. Standard ^1^H and ^13^C NMR spectra for 4-phenylthiopicolinonitrile and 4-phenylthiopicolinohydrazonamide in DMSO-*d*_6_ were recorded on a Varian Unity Plus instrument (Palo Alto, CA). IR spectra (KBr) were determined as KBr pellets of the solids on a Satellite FT-IR spectrophotometer (Madison, WI). Melting points were determined via a Stuart SMP30 apparatus and were retained without any corrections.

#### 2.1.1. 4-Phenylthiopicolinonitrile

First, 5.60 g (40 mmol) of 4-chloropicolinonitrile and 4.91 mL (48 mmol) of thiophenol were dissolved in 25 mL of dioxane, and then 6 mL (40 mmol) of DBU (1,8-diazabicyclo [5.4.0]undec-7-ene) were added. The mixture was refluxed for 1 h. Next, some of the solvent was evaporated and ice was added. The resulting precipitate was filtered off and dried. The crude product was purified by recrystallization from cyclohexane. Yield: 98%; mp 51–53 °C; IR (KBr): 3047 (υ CAr-H); 2234 (υ C ≡ N); 1647 (υ C = N); 1570, 1437 (υ C = C); 835, 759, 691 (γ C–H) cm^−1^; ^1^H NMR (500 MHz, DMSO-*d_6_*): δ 7.19 (dd, 1H, pyridine, J_1_ = 2 Hz, J_2_ = 3 Hz); 7.54-7.63 (m, 5H, ArH); 7.68 (d, 1H, pyridine, J = 2Hz); 8.47 (d, 1H, pyridine, J = 5Hz) ppm; Elemental analysis for C_12_H_8_N_2_S (212.04) calculated: C, 67.90; H, 3.80; N, 13.20; found: C, 67.73; H, 3.79; N, 13.02.

#### 2.1.2. 4-Phenylthiopicolinohydrazonamide

To a solution of 2.12 g (10 mmol) of 4-phenylthiopicolinonitrile in 75 mL of methanol, 2 mL (13.5 mmol) of DBU was added. The mixture was refluxed for 4 h. Next, some of the solvent was evaporated, and 1 mL (31.7 mmol) of an 80% hydrazine hydrate solution was added. The solution was refluxed for 15 min. The mixture was then cooled, and 5 mL of diethyl ether was added. The precipitate was filtered off, dried and recrystallized from methanol. Yield: 65%; mp 101–103 °C; IR (KBr): 3445, 3351 (υ N-H); 3072, 3051 (υ CArH); 1574 (δ N-H); 1469, 1439 (υ C = C); 750, 689 (γ C-H) cm^−1^; ^1^H NMR (500 MHz, DMSO-d_6_): δ 6.46 (br s, 2H, NH_2_); 7.17 (dd, 1H, pyridine, J_1_ = 2Hz, J_2_ = 3Hz); 7.50–7.58 (m, 4H, 2ArH + NH_2_); 7.61–7.63 (m, 3H, ArH); 7.73 (d, 1H, pyridine, J = 2Hz); 8.40 (d, 1H, pyridine, J = 5Hz) ppm; ^13^C NMR (175 MHz, DMSO-d_6_): δ 117.27; 121.94; 128.75; 130.57; 130.75 (2C); 130.89; 135.59; 135.66 (2C); 148.72; 150.53 ppm; Elemental analysis for C_12_H_12_N_4_S (244.08) calculated: C, 58.99; H, 4.95; N, 22.93; found: C, 59.04; H, 4.93; N, 22.81.

#### 2.1.3. 2-[Amino-(4-phenylthiopyridin-2-yl)methylene]-N-cyclohexylhydrazinecarbothioamide (**1**)

First, 0.244 g (1 mmol) of 4-phenylthiopicolinohydrazonamide was dissolved in 5 mL of dioxane, and then 0.155 mL (1.1 mmol) of cyclohexyl isothiocyanate was added. The mixture was refluxed for 5 min. Next, the mixture was cooled, and diethyl ether was added. The product was filtered off, dried and recrystallized from ethanol. Yield: 58%; mp 200–202 °C; IR (KBr): 3398, 3343, 3172 (υ N–H); 2929, 2851 (υ C–H); 1658 (υ C = N); 1573, 1527 (δ N–H); 1220 (υ C–N); 750 (γ Car–H) cm^−1^; ^1^H NMR (500 MHz, DMSO-*d_6_*): δ 1.17–1.29 (m, 5H, CH); 1.61–1.84 (m, 5H, CH); 4,05 (s, 1H, CH); 6.91 (s, 2H, NH_2_); 7.11 (dd, 1H, pyridine, J_1_ =2 Hz, J_2_=3 Hz); 7.26 (d, 1H, ArH, J = 9 Hz); 7.54–7.55 (m, 3H, ArH); 7.62–7.64 (m, 2H, ArH +NH); 7.71 (s, 1H, pyridine); 8.35 (d, 1H, pyridine, J = 5 Hz); 9.97 (s, 1H, NH) ppm; Elemental analysis for C_19_H_23_N_5_S_2_ (385.14) calculated: C, 59.19; H, 6.01; N, 18.16; found: C, 58.82; H, 5.75; N, 17.87.

### 2.2. Tuberculostatic Activity

Compounds **1**–**6** were tested using the classic test tube method consisting of successive dilution in the Proskauer and Beck modification on Youmans liquid medium with the addition of 10% bovine serum [15,16]. Bacterial suspensions were prepared from 14-day-old cultures of slow-growing strains and 48-h cultures of saprophytic strains [17]. The obtained compounds were dissolved in ethylene glycol at a concentration of 10 mg mL. Next, exponential dilutions were made in Youmans medium. Incubations were carried out at 37 °C for 21 days. The minimum inhibitory concentration (MIC) values were defined as the minimum concentration that inhibits the growth of the tuberculosis strains tested. As a control, a medium containing no test compounds was used. Isoniazid (INH) and pyrazinamide (PZA) were used as reference drugs.

### 2.3. X-ray

The crystal measurements of compounds **1**, **2**, **5** and **6** were performed on an XtaLAB Synergy diffractometer, Dualflex, Pilatus 300K (Rigaku Corporation, Tokyo, Japan), while measurements of compounds **3** and **4** were performed on Bruker SMART APEXII CCD Diffractometer (Bruker AXS Inc., Madison, WI, USA). All diffraction experiments were carried out with CuKα radiation. For data processing of compounds **1**, **2**, **5** and **6,** CrysAlis PRO (Rigaku Oxford Diffraction Ltd., Yarnton, Oxfordshire, UK, 2020) was used. The diffraction data of compounds **3** and **4** were processed with SAINT ver. 8.34A, SADABS ver. 2014/4 and XPREP ver. 2014/2 (Bruker AXS Inc., Madison, WI, USA). The structures were determined with the ShelXT 2018/2 solution program (Version 2018/2, 2018, Göttingen, Germany) [18] and refined with ShelXL 2018/3 [19] using full matrix least-squares minimization on F2. For visualization, ShelXle [20] was used. All H atoms (except those engaged in hydrogen bonds) were geometrically optimized and allowed as riding atoms, with the appropriate C—H distances for the respective temperatures of the diffraction experiments and with Uiso(H) = 1.2 Ueq(C, N). All studied crystals were obtained by the slow evaporation of the solvent, a mixture of methanol and DMF (1:1). The tables and figures were prepared using the publCIF [21] and Mercury [22] programs.

CCDC 2125507, 2125509, 2125514, 2125531, 2125534, 2125535 contain the supplementary crystallographic data for this paper. The data is provided free of charge by The Cambridge Crystallographic Data Centre via www.ccdc.cam.ac.uk/structures.

### 2.4. Solution NMR

All NMR experiments were run at 298 K on a 600 MHz Bruker Avance III spectrometer, equipped with an HCN probe head operating at 600.15 and 60.81 MHz for ^1^H and ^15^N nuclei, respectively. The samples were prepared in DMSO-d6 (99.8% + D) from Armar Chemicals. The ^13^N NMR data were assigned using standard 2D 1H-^15^N NMR correlation techniques and gradient-selected heteronuclear single-quantum correlation (gs-HSQC) [23].

### 2.5. Solid-State NMR

Solid-state cross-polarization magic angle spinning (CP/MAS) NMR experiments were performed on a 400 MHz Avance III spectrometer (operating at 400.13 MHz, 40.55 MHz for ^1^H and ^15^N) equipped with a MAS probe head using 4-mm ZrO_2_ rotors. A sample of ^13^C, ^15^N-labeled histidine hydrochloride was used to set the Hartmann–Hahn condition and the secondary reference sample for ^15^N. The ^15^N CP/MAS spectra were performed with a proton 90° pulse length of 4 μs, a contact time of 2 ms, a repetition delay of 25 s and 5 s for compounds **1** and **4**, respectively. A spectral width of 25 kHz and a time-domain size of 32 k data points were applied. The acquisition data were collected with a SPINAL decoupling sequence [24]. Spectra were processed on a PC using the Bruker TopSpin 3.6 program [25].

### 2.6. QM Calculations

The DFT calculations were performed using the CASTEP 19.11 code [26]. The geometry optimization of all atomic positions and unit cell parameters were performed, starting with the X-ray diffraction crystal structures of compounds **1** and **4**. The PBE (Perdew–Burke-Ernzerhof) functional with the TS (Tkatchenko-Scheffler) dispersion correction scheme (DFT-D method) was applied [27,28,29]. A comparison of the average forces remaining on the atoms after geometry optimization was carried out using a maximum plane-wave cutoff energy of 800 eV and ultrasoft pseudopotential [30]. A Monkhorst-Pack grid was used to sample the Brillouin zone [31]. The NMR chemical shifts were computed using the gauge, including the projected augmented wave (GIPAW) method [32,33]. In all cases, the optimization algorithm was Broyden–Fletcher–Shanno-Goldfarb (BFSG ) with a line search [34].

## 3. Results and Discussion

### 3.1. Synthesis

The new derivative 2-[amino-(4-phenylthiopyridin-2-yl)methylene]-*N*-cyclohexylhydrazinecarbothioamide **1** was obtained according to the presented Figure 1. The substrate for the synthesis was commercially available 4-chloropicolinonitrile (Apollo Scientific, Stockport, UK). In the first step, the chlorine atom in the C4 position was subjected to a thiophenyl substitution reaction. The reaction was carried out in dioxane with the addition of 1,8-diazabicyclo [5.4.0]undec-7-ene (DBU). To prepare the hydrazonamide derivative of 4-phenylthiopicolinonitrile, an 80% solution of hydrazine hydrate was added to the methyliminoester solution prepared in situ. A method developed at the Department of Organic Chemistry, Medical University of Gdańsk was used to obtain the methylimino ester, which consisted of carrying out the reaction in a methanol environment and DBU. The reaction was carried out for 4 h in reflux. The obtained 4-phenylthiopicolinohydrazonamide was reacted with cyclohexyl isothiocyanate according to the method described by Brown et al. [35]. The yields of the performed reactions were in the range of 58–98%. The synthesis of compounds **2**–**6** was described by the authors earlier [14].

### 3.2. Tuberculostatic Activity

The obtained compounds were tested in vitro for their tuberculostatic activity. The study was carried out on two strains of *M. tuberculosis*: standard H37Rv and a clinical strain isolated from patients of Spec. 210, which showed resistance to isoniazid, rifampicin, ethambutol and *p*-aminosalicylic acid. Their anti-mycobacterial activity was expressed by the minimum inhibitory concentration (MIC) defined as the minimum concentration inhibiting the growth of the tested microorganisms. Significant discrepancies in the MIC and its low value in all tested compounds prompted us to study their structure in more detail.

All compounds tested, except compounds **1** and **6**, showed moderate antitubercular activity with MIC values ranging from 6.2–12.5 µg/mL [14]. Compound **1**, possessing a cyclohexyl ring at a terminal N atom and a hydrogen atom at this nitrogen, showed the lowest activity (MIC 25–50 µg/mL). Moreover, this compound showed two times lower activity against the resistant strain, while all other compounds presented in Table 1 did not show different activity against the resistant and the standard strains. As only **1** appeared in neutral form, while all other compounds studied here (having higher activity) were detected as zwitterions in the solid-state, one may suggest that the latter form is “an active one”. Among the studied species existing in zwitterionic form, compound **6** showed the highest activity with a MIC of only 0.4 µg/mL, most probably due to a good fit with an unknown receptor. Moreover, the compounds demonstrated the same antitubercular activity against the standard H37Rv as against the resistant Spec. 210 strain. Compared with the values ascribed to isoniazid (MIC of 3.1 and 12.5 µg/mL), the values for **6** are almost eight times less for the standard strain and as much as 31 times less (**6** being more active), respectively, for the resistant bacteria.

### 3.3. Crystallography 

The crystal structures of six compounds were determined (Table 2, Figure 3). Only compound **1** adopts the neutral form, while the others form zwitterions in the solid-state.

The structure of compound **1** is the only one in which molecules form dimers through strong hydrogen bonds of the N—H···N type (Table 3, Figure 4). These dimers form ribbons by hydrogen bonds of the N—H···S type. The conformation of the molecule is stabilized by intramolecular hydrogen bonds (N4—H4···N42). The molecule fragment (*N*’-thioformylpicolinohydrazonamide) takes a flat form (r.m.s. deviation = 0.1586 and 0.0762 Å). Interestingly, DFT calculations of energies (for **6**-311 G(d,p) standard basis set and taking into account the solvent effect of water using the polarizable continuum model) for the optimized geometries of molecules **1** and **4** revealed that the forms present in the crystals are more stable than the corresponding tautomeric forms, e.g., the energy of the neutral form of **1** has an energy lower by about 8 kcal/mol, and the energy of both molecules of **4** is lower in the zwitterionic form by approximately 4 kcal/mol.

Contrary to compound **1**, in a crystal state, the molecules of all other compounds studied here adopt the form of zwitterions with hydrogen found at the N3 atom. The pyridine ring in these structures is in a reversed position and forms characteristic bifurcated intramolecular hydrogen bonds N(42)···H(N3)···S1 (Table 4, Table 5, Table 6, Table 7 and Table 8). Additionally, rotation along the N2-C1 bond occurs.

Structure **2** contains two molecules in the independent part of the unit cell. The molecules form infinite chains through N-H···S hydrogen bonds (Table 4, Figure 5). The conformation of the molecules is stabilized by intramolecular hydrogen bonds of the N-H···N type. The molecule fragment (*N*’-thioformylpicolinohydrazonamide) takes a flat form (r.m.s. deviation = 0.0421 and 0.0846 Å).

In its crystal, the molecules of compound **3** form a chain via N4—HbB···S1 type hydrogen bonds (Figure 6, Table 5). Additionally, intramolecular hydrogen bonds are formed (N4—H4A···N2). The molecule fragment (*N*’-thioformylpicolinohydrazonamide) takes a flat form (r.m.s. deviation = 0.0847 Å).

In the structure of compound **4**, there are two molecules of the compound, two molecules of water and two molecules of DMF (one disordered), in the independent part of the unit cell. The molecules have different positions of substituents in the piperazine ring (equatorial–equatorial or equatorial–axial), with the phenyl always in the equatorial position. This compound also forms chains of hydrogen-bonded molecules with the water molecules as “bridges” (Table 6, Figure 7). The DMF molecules fill the gaps and form hydrogen bonds of the N-H···O type (Figure 8). The molecule fragment (*N*’-thioformylpicolinohydrazonamide) takes a flat form (r.m.s. deviation = 0.0655 and 0.0830 Å).

The molecules of **5** form chains via N3-H3···S1 hydrogen bonds (Table 7, Figure 9) in the studied crystal. Additionally, intramolecular hydrogen bonds (N4-H4B···S1) are formed. The molecule fragment (*N*’-thioformylpicolinohydrazonamide) takes a flat form (r.m.s. deviation = 0.0775 Å).

The crystal structure of compound **6** reveals the standard N-H···S hydrogen bonds as the rest of the zwitterion compounds studied here, forming infinite chains (Table 8, Figure 10). Additionally, the molecules form intermolecular hydrogen bonds with oxygen atoms in morpholine rings. The molecule fragment (*N*’-thioformylpicolinohydrazonamide) takes a flat form (r.m.s. deviation = 0.0871 Å).

The molecules of compound **6** in the zwitterion form have an extended conformation, while compound **1** (neutral form) has a bent form (Figure 11). The pattern of intramolecular hydrogen bonds depends on the form adopted.

There are 16 analogous crystal structures in the CSD [36,37] database: eight are in the zwitterionic form, and eight are in the neutral one (Figure 12). Several of them originated from our laboratories, and their antitubercular activity is known. Unfortunately, other studies on the crystal structure of similar compounds do not mention their activity, except that of Almeida et al. [38], which quotes the lowest MIC of about 160 [µg/mL] for antitubercular action, well above the values found in the studied species here (Table 1). All compounds in each of the two groups have the same intramolecular pattern of hydrogen bonds related to the form in which the molecules occur, which is shown (Figure 12 and Figure 13) by overlaying an N(H_2_)-C-N-N-C(S) fragment in all 16 structures. Only compounds in zwitterionic form adopt an elongated geometry.

### 3.4. NMR

The ^15^N CP MAS spectra of compound **1** (a) and compound **4** (b) are shown in Figure 14. As described in the previous section, both structures contain two crystallographically nonequivalent molecules in the asymmetric part of the unit cell. It is also easily visible for the compound **1** spectrum, where most of the positions show separated resonance lines. However, even a brief look at the compound **4** spectrum shows that most of the positions are isochronous and overlap, giving single resonances. The full assignment of ^15^N CP MAS signals to the conformers is not an easy task. To solve this problem, we applied the GIPAW method. The usefulness of this approach for the assignment of solid-state NMR signals is unquestionable [48,49,50]. After comparing the experimentally obtained chemical shifts with GIPAW calculation nuclear shieldings, we were able to assign ^15^N CP MAS signals, as shown in Figure 14. The numerical values are presented in Table 9 and Table 10.

The main question regarding these two structures was the localisation of the hydrogen atom in the -N = NH- link. These positions are indicated as N2 and N3 in the crystal structures for both structures. Keeping in mind the uncertain location of the hydrogen atom we investigated, two theoretical models with the different locations of the hydrogen atom were constructed, and GIPAW calculations were performed. Our results, presented in Table 9 and Table 10, fully supported the supposition that in a solid-state, the hydrogen atom is localised in the same position where it was found in single crystal X-ray structures. However, it has to be stressed that such a statement is not fully unambiguous for compound **1** since the differences between both tautomers are rather small. Nevertheless, it means that the selected single crystal for X-ray measurement is representative of the bulk material. Our additional ^15^N NMR measurements in a DMSO-d_6_ solution provided the same conclusion about the studied material. The ^15^N NMR spectra clearly show only one cross peak in Figure 15, which suggests that there is only one tautomeric form, also in the liquid state. The chemical shifts recorded for the –NH = signal are very similar to the solid-state NMR values, which suggests that the same tautomeric form is in the solid-state and the DMSO-d_6_ solution.

## 4. Conclusions

The structures of six new 2-amidinothiosemicarbazone derivatives of pyridine were determined in the solid-state by means of X-ray diffraction and NMR. For the first time, synthesis and tuberculostatic activity are also presented for compound **1**. The studied amidrazone derivatives exist either in zwitterionic or in neutral form. Zwitterionic molecules in the solid-state adopt an extended conformation due to conjugation and a unique intramolecular hydrogen-bonds pattern, which secures both the planarity and similarity of the common frame of these molecules. Protonated nitrogen at the C = S group, i.e., having other properties, may influence the preferred form of the molecule. In general, the compounds that prefer the zwitterionic state in their crystals show higher antitubercular activity than the compound detected in the neutral form.

## Data Availability

Data are contained within the article.

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
