# Peer review of "Relationship between the Crystal Structure and Tuberculostatic Activity of Some 2-Amidinothiosemicarbazone Derivatives of Pyridine"

_materials, 2022, doi:10.3390/ma15010349_

Round 1
Reviewer 1 Report
See in attached pdf file

Reviewer 2 Report
This paper describes the solid state (mainly RX) structures of a number of putative tuberculostatic compounds. The study is well conducted.
Normally, to discuss about planar forms, it is useful to characterise either plane angles or "pseudo-chiral" volumes (i.e. in resonant systems = 2π) in addition to H bonds. Nothing is reported here to differentiate compound (1) from the others on that basis. Not easy when looking at fig. 8-11 to discriminate which is which.
Note also that in the refinements at 100 K (cryo temp.) the imposed C-H distance should not be equal to 0.95 Å... but shorter.
I have no particular additional remarks except that wording should be carefully inspected before publication:
-line 39 : .. .. has access to ...
-qoutes line 306, also compund lines 332,
line 352 nevertheless, show line 355 etc ...
Re-arrange the first column (hydrogen on, (hydrogen...?? in Tables 9 (line 338) and 10 (line 342),
-Molecules extracted from CSD: give acronyms, if possible.
-In the reference list:
-Font size : line 399,
-Check volume numb. (italic/standard) and pages in references lines 396, 402, 456
-lines 420, : Bacteriol.
-Adopt the same abbreviations : J. Appl. Cryst. or J. Appl. Crystallogr. (lines 425, 427, 430 v.s. 458, 459)
Reviewer 3 Report
Suggestions for Authors are in the attached pdf file

Round 2
Reviewer 3 Report
The authors took into account all the comments of the reviewer and tried to explain the controversial points. The article can be published in its present form.